# Molecular Epidemiology of Carbapenem-Resistant *Acinetobacter baumannii* Strains Isolated at the German Military Field Laboratory in Mazar-e Sharif, Afghanistan

**DOI:** 10.3390/microorganisms9112229

**Published:** 2021-10-26

**Authors:** Paul G. Higgins, Meret Kniel, Sandra Rojak, Carsten Balczun, Holger Rohde, Hagen Frickmann, Ralf Matthias Hagen

**Affiliations:** 1Institute for Medical Microbiology, Immunology, and Hygiene, Faculty of Medicine and University Hospital Cologne, University of Cologne, 50935 Cologne, Germany; paul.higgins@uni-koeln.de; 2German Centre for Infection Research (DZIF), Partner Site Bonn-Cologne, 50935 Cologne, Germany; 3Department of Microbiology and Hospital Hygiene, Bundeswehr Central Hospital Koblenz, 56070 Koblenz, Germany; meret.kniel@web.de (M.K.); sandrarojak@bundeswehr.org (S.R.); carsten1balczun@bundeswehr.org (C.B.); 4Institute of Medical Microbiology, Virology and Hygiene, University Medical Center Hamburg-Eppendorf (UKE), 20251 Hamburg, Germany; rohde@uke.de; 5Department of Microbiology and Hospital Hygiene, Bundeswehr Hospital Hamburg, 20359 Hamburg, Germany; frickmann@bnitm.de or; 6Institute for Medical Microbiology, Virology and Hygiene, University Medicine Rostock, 18057 Rostock, Germany

**Keywords:** *Acinetobacter baumannii*, molecular epidemiology, carbapenem resistance, Afghanistan, surveillance

## Abstract

The study was performed to provide an overview of the molecular epidemiology of carbapenem-resistant *Acinetobacter baumannii* in Afghanistan isolated by the German military medical service during the Afghanistan conflict. A total of 18 isolates were collected between 2012 and 2018 at the microbiological laboratory of the field hospital in Camp Marmal near Mazar-e Sharif, Afghanistan, from Afghan patients. The isolates were subjected to phenotypic and genotypic differentiation and antimicrobial susceptibility testing as well as to a core genome multi-locus sequence typing (cgMLST) approach based on whole-genome next-generation sequence (wgNGS) data. Next to several sporadic isolates, four transmission clusters comprising strains from the international clonal lineages IC1, IC2, and IC9 were identified. Acquired carbapenem resistance was due to *bla*_OXA-23_ in 17/18 isolates, while genes mediating resistance against sulfonamides, macrolides, tetracyclines, and aminoglycosides were frequently identified as well. In conclusion, the assessment confirmed both the frequent occurrence of *A. baumannii* associated with outbreak events and a variety of different clones in Afghanistan. The fact that acquired carbapenem resistance was almost exclusively associated with *bla*_OXA-23_ may facilitate molecular resistance screening based on rapid molecular assays targeting this resistance determinant.

## 1. Introduction

Carbapenem-resistant *Acinetobacter baumannii* has been identified as a considerable menace for traumatic wounds associated with international wars and crises [1,2]. This also applies to the recent military conflict in Afghanistan, in which German soldiers participated for nearly 20 years, from 2002 until 2021.

Most reports on war trauma-associated *A. baumannii* infections or colonization from the recent Afghanistan conflict were, however, provided by the medical services of the USA and Canada [3,4,5,6,7,8,9,10,11]. As early as 2004, a report on a limited number of *A. baumannii*-associated bloodstream infections associated with Operation Enduring Freedom (OEF) in Afghanistan was published [3]. In 2006, carbapenem-resistant isolates were still reported to be acquired predominantly in Iraq and less frequently in Afghanistan [4]. One year later, however, Canadian health officials published warnings about the risk of spreading multidrug-resistant *A. baumannii* strains imported into Canadian hospitals from Afghan battlefields [5]. Subsequently, in 2008, the US Armed Forces Medical Service reported an increased frequency of *A. baumannii* isolates associated with osteomyelitis in war-injured soldiers fighting for the OEF mission [6], and in the same year, an increased acquisition risk of resistant *A. baumannii* isolates at OEF deployment sites was documented [7]. Together with *Staphylococcus aureus* and *Klebsiella pneumoniae*, *A. baumannii* accounted for nosocomial infection rates of 2–4% in injured patients from the OEF mission [8], and a risk of spreading resistance was identified. However, as reported in 2011, the colonization rate with *A. baumannii* was highest at the beginning of the mission and declined in the following years [9]. Nevertheless, the relevance of resistant or even multidrug-resistant *A. baumannii* strains in the wounds of war-injured US trauma patients remained considerable, accounting for 38.6% of the registered infections due to multidrug-resistant Gram-negative bacteria [10]. In a recent study on trauma-related infections in US soldiers, *Acinetobacter* spp. were isolated from 8.7% of skin and soft tissue infections, 9.0% of bloodstream infections, and 11.0% of osteomyelitis cases, [11]. Consequently, research on substances potentially inactivating *A. baumannii* in traumatic war injuries is ongoing [12].

German soldiers in Afghanistan were affected by multidrug-resistant *A. baumannii* as well, but the documentation is considerably less comprehensive. In 2011, Vandersee and colleagues [13] started reporting antimicrobial resistance issues associated with bacterial pathogens isolated from deployed German soldiers as well as from local allies in Afghanistan, which was later confirmed by Helm in 2013 [14].

Unfortunately, however, little published information is available on the molecular epidemiology of Afghan *A. baumannii* isolates or their antibiotic resistance determinants, although respective typing efforts have been described since 2006 [4]. In the meantime, 27,000 bacterial strains have been sequenced in the course of the Antimicrobial Resistance Monitoring and Research (ARMoR) program at the Walter Reed Army Institute of Research (Silver Spring, MD, USA) [15]. Nevertheless, available information remains scarce. In a recent Swedish assessment, the predominance of *A. baumannii* international clone 1 (IC1) was reported for Afghanistan, with IC1 lineage 3 being virtually limited to Afghanistan, Pakistan, and India [16]. In a recent small study [17], four carbapenem-resistant *A. baumannii* were included, which were isolated in the course of a minor outbreak in a German field hospital in northern Afghanistan in 2008; shipped to the US Regional Medical Center Landstuhl (RMCL), Germany; and donated by RMCL to the Bundeswehr Hospital Hamburg, Germany, for a study on fluorescence-in-situ-hybridization (FISH)-based identification of *Acinetobacter* spp. in 2011 [18]. Those four *A. baumannii* strains were identified as international clone 1 (IC1), Oxford sequence type (ST) 498, and Pasteur ST 81 [19,20,21] and carried the beta-lactamase genes *bla*_ADC-25__-like_*, bla*_OXA-23_*,* and *bla*_OXA-69_, as well as the genes *aadB*-like, *aph*(*3*′)-*Ia,* and *aph*(*3*′)-*VIa*-like, mediating aminoglycoside resistance, next to *sul*-*2* which mediates sulphonamide resistance [17].

Beginning in 2012, multidrug-resistant bacterial strains isolated at the microbiological laboratory of the German military field hospital in Camp Marmal near Mazar-e Sharif, Afghanistan, were sent to the laboratory at the Bundeswehr Central Hospital in Koblenz, Germany. Within a period of 6 years from 2012 to 2018, a total of 18 carbapenem-resistant *A. baumannii* strains isolated from Afghan patients were shipped from Mazar-e Sharif, Afghanistan, to Koblenz, Germany. In the study presented here, those strains were assessed by whole-genome next-generation sequencing (wgNGS) for clonal relationship testing, and phenotypic resistance testing was compared to identify genetic resistance determinants. By doing so, the study intended to contribute to the so-far scarcely available knowledge on the local epidemiology of carbapenem-resistant *A. baumannii* in Afghanistan.

## 2. Materials and Methods

### 2.1. Strains

A total of 18 carbapenem-resistant *Acinetobacter baumannii* strains were isolated from both clinical and screening samples of Afghan patients (*n* = 18) at the microbiological laboratory of the German military field hospital at Camp Marmal near Mazar-e Sharif, Afghanistan, between the years 2012 and 2018. After isolation, the strains were transferred to the microbiological laboratory of the Bundeswehr Central Hospital in Koblenz, Germany, for confirmatory testing as well as surveillance purposes. To ensure the patients’ anonymity, patient-specific details are not provided in agreement with the ethical clearance for this study.

### 2.2. Phenotypic Assessments

At the Bundeswehr Central Hospital in Koblenz, Germany, a preliminary confirmation of the species identities at the *A. baumannii* complex level was performed by matrix-assisted laser-desorption-ionization time-of-flight mass spectrometry (MALDI TOF MS) using a Bruker MALDI Biotyper^TM^ MBT^TM^ smart mass spectrometer and the software MBT compass version 4.1. Antimicrobial resistance was assessed using N 248-cards of a VITEK-II automate (BioMérieux, Marcy-l’Étoile, France). Because of the known problem of poor reliability of VITEK-II-based colistin resistance testing [22], the microbroth dilution assay MICRONAUT-S (MERLIN Diagnostika GmbH, Bornheim, Germany) was applied for this purpose. Phenotypic resistance testing results were interpreted according to the European Committee on Antimicrobial Susceptibility Testing (EUCAST) standards version 11.0.

### 2.3. Genotypic Assessments

From all assessed strains, total DNA was extracted using the DNeasy Ultra Clean Microbial Kit (Qiagen, Hilden, Germany) as suggested by the manufacturer and shipped to the Institute for Medical Microbiology, Immunology, and Hygiene of the University of Cologne, Germany, for further molecular assessments. In detail, the species-identity of the *A. baumannii* strains was confirmed by *gyrB* multiplex PCR [23,24]. Additional screening PCRs [25,26] targeting common beta-lactamases were applied. In addition, whole-genome sequencing from the extracted nucleic acids of the strains was performed as previously described with minor modifications [17,27]. Briefly, sequencing libraries were prepared with the purified DNA using the Ultra II FS DNA Library Prep Kit (New England Biolabs, Frankfurt, Germany) for 250 bp paired-end sequencing runs on an Illumina MiSeq sequencer. Subsequently, the Velvet assembler integrated with the Ridom SeqSphere+ v.7.2.3 software performed de novo assembling of the obtained reads. Project-associated raw sequencing reads were submitted to the European Nucleotide Archive (https://www.ebi.ac.uk/ena/ (accessed on 22 October 2021)) with the accession number PRJEB47537. The sequences were subjected to a validated core genome multi-locus sequence typing (cgMLST) scheme [28] using the Ridom SeqSphere^+^ v. 7.2.3 software (Ridom GmbH, Münster, Germany). By generating a minimum spanning tree including 2390 target alleles using the same software tool, also including an in-house library with the established international clones, the clonal relationship between the isolates was visualized. In addition, the genome assemblies of all isolates were used to identify sequence types (STs) according to the Oxford and the Pasteur 7-loci MLST schemes [19,20,21], as shown on the pubMLST website (https://pubmlst.org/abaumannii/ (accessed on 22 October 2021)). The acquired resistome was identified using the webtool ResFinder 2.1 (http://www.bldb.eu/ (accessed on 22 October 2021)) (batch upload), and *bla*_OXA_ genes were identified using the BLAST function of the beta-lactamase database (http://www.bldb.eu/ (accessed on 22 October 2021)) [29,30].

### 2.4. Ethical Clearance

Ethical clearance for the anonymized retrospective assessment of the strains without the requirement of informed consent was obtained from the ethics committee of the medical association of Rheinland-Pfalz (reference number 2021-16003, provided on 30 July 2021). The study was conducted in line with the requirements of the Declaration of Helsinki.

## 3. Results

### 3.1. Phylogeny Based on Core Genome Sequence Typing

The cgMLST analysis indicated four transmission clusters, comprising six, four, two, and two isolates, respectively (Figure 1). Three out of those four transmission clusters could be attributed to the international clones (IC) IC1, IC2, and IC9, while the other cluster was genetically distinct from the international clonal lineages. The remaining four isolates were singletons, although three of them clustered within the international clonal lineages but without sufficient genomic similarity to suggest nosocomial transmission (Figure 1).

In addition, the sequence assemblies of all strains were subjected to seven-loci multi-locus sequence typing (MLST) schemes, i.e., the Oxford scheme and the Pasteur scheme (Table 1) [19,20,21]. Isolates clustering with the ICs had the expected Pasteur STs. With one exception, IC1 and IC2 isolates had two Oxford STs owing to their double *gdhB* loci.

### 3.2. Genotypic and Phenotypic Resistance

Genotypic resistance testing identified multiple genetic determinants. In all isolates, carbapenem-resistance was associated with an acquired OXA-type gene, with *bla*_OXA-23_ found in all except one isolate, which had *bla*_OXA-58_. One isolate had an additional *bla*_GES-11_. All isolates possessed the intrinsic *bla*_ADC_ (not shown in Table 2 below) and *bla*_OXA-51-like_, with variants associated with the international clones. Multiple acquired aminoglycoside modifying enzymes were associated with aminoglycoside resistance, comprising *aadB*-like, *aac*(*3*)-*Ia*-like, *aacA4*-like, *aadA1*, *aph*(*3*′)-*Ic*, *aph*(*3*′)-*Ic*-like, *aac*(*3*)-*IId*-like, *aph*(*3*′)-*VIa*-like, *armA*, *strA*, and *strB* (Table 2). High gentamicin minimum inhibitory concentrations (MICs) were associated with *aadB*-*like*, while amikacin resistance was associated with *armA* in one isolate and *aph*(*3*′)-*VIa* in ten isolates. However, two isolates with this gene were amikacin-susceptible. The element *aac*(*6*′)*Ib*-*cr*-like identified in one strain is linked to combined aminoglycoside and fluoroquinolone resistance. However, fluoroquinolone resistance was associated with target-site mutations in the *gyrA* and *parC* genes. Further, the *tet*-gene variants *tet*(*39*), *tet*(*B*), and *tet*(*B*)-like, mediating tetracycline resistance; *sul1* and *sul2*, mediating sulphonamide resistance; and *mph*(*E*) and *msr*(*E*), mediating macrolide resistance, were among the more frequently observed resistance determinants. By contrast, *ARR*-*2*, mediating rifampicin resistance; *cmlA1*-like, mediating phenicole resistance; and *dfrA7*, mediating trimethoprim resistance, were only recorded once each. Details are provided in Table 2, and the corresponding phenotypic resistance details are summarized in Table 3 below.

## 4. Discussion

This study was performed to provide information on the molecular epidemiology of carbapenem-resistant *A. baumannii* strains collected from patients in Afghanistan by deployed soldiers of the German military medical service in the course of the recent conflict in Afghanistan. The assessment identified strains that clustered with the international clonal lineages IC1, IC2, and IC9 next to several singletons. The regional detection of IC1 strains aligns well with previous reports [16,17], while IC2 was recently reported to be prevalent in civil war-injured Syrian patients [17]. The MLST types observed for the IC1 isolates, however, were different from the previously reported ones [17], and the cgMLST analysis indicated a distance of 619 alleles between the IC1 cluster from the previous assessment [17] and the present one. Therefore, the study suggests that even within IC1, considerable phylogenetic distance can be expected for *A. baumannii* clones prevalent in Afghanistan. By contrast, the IC2 isolates from this study were only 92 alleles distant from an IC2 isolate from Syria [17].

As four transmission clusters and only a few singletons were observed, this assessment confirms previous observations by the German military medical service that multidrug-resistant *A. baumannii* occurred predominantly associated with outbreak scenarios in Afghanistan in military medical infrastructure [13,14], while randomly isolated strains apart from outbreak events were quite rare. Unfortunately, the retrospective design of the study and the ethical need for anonymization prevented more detailed assessments of likely transmission chains.

In line with previous findings [17], carbapenem resistance was nearly exclusively associated with the abundance of the *bla*_OXA-23_ gene, while various genes mediating resistance against sulfonamides, macrolides, tetracyclines, and aminoglycosides were also identified. The nearly monogenetic association of carbapenem-resistance in the assessed Afghan *A. baumannii* isolates may be of practical interest if PCR-based screening based on automated point-of-care testing (POCT) assays for carbapenem-resistant *A. baumannii* in Afghan patients are considered for hygiene reasons or guidance of antimicrobial therapy [31].

Focusing on residual susceptibility toward other tested antibiotic drugs, ubiquitous susceptibility toward colistin could be demonstrated next to heterogeneous susceptibility toward the aminoglycosides amikacin and tobramycin. Despite the abundance of aminoglycoside modifying enzymes, with all but one isolate in possession of ≥2 resistance genes, two isolates were susceptible to tobramycin, and seven were susceptible to amikacin, while all were gentamicin-resistant. This highlights that it is not always possible to correlate phenotype with genotype and emphasizes that minimum inhibitory concentration (MIC) testing cannot easily be replaced by molecular resistance prediction. Furthermore, efflux can also contribute to aminoglycoside resistance in *A. baumannii*, but we did not investigate this in this study. Of note, tigecycline was not assessed, as susceptibility testing results were interpreted in line with EUCAST (European Committee on Antimicrobial Susceptibility Testing) standards, which do not describe breakpoints for tigecycline.

The study has a number of limitations. First, the available number of isolates was quite low, limiting the representativeness of the assessment. Second, and for the same reason, the anonymity of the patients demanded abstaining from a very detailed presentation of clinical information. In some cases, these data were also incompletely documented and thus only partly available. Third, the completeness of the collection is uncertain, as only isolates transferred from Afghanistan could be included in the sample collection in Koblenz. Accordingly, the results can only provide another piece in the puzzle of the molecular epidemiology of carbapenem-resistant *A. baumannii* strains in Afghanistan. Fourth, no long-read sequencing was performed in the course of the present study. Therefore, no information on the exact positions of insertion sequences in the bacterial genomes and thus on their functional relevance can be provided.

## 5. Conclusions

In spite of the above-mentioned limitations, the assessment provided a number of interesting insights. Therefore, the abundance of the international clonal lineages IC1, IC2, and IC9 of *A. baumannii* next to several out-standers could be confirmed for Afghanistan. Further, it was shown that acquired carbapenem-resistance in the assessed Afghan *A. baumannii* strains was nearly exclusively due to *bla*_OXA-23_ genes, which may be a helpful hint for molecular diagnostic purposes.

## Figures and Tables

**Figure 1 microorganisms-09-02229-f001:**
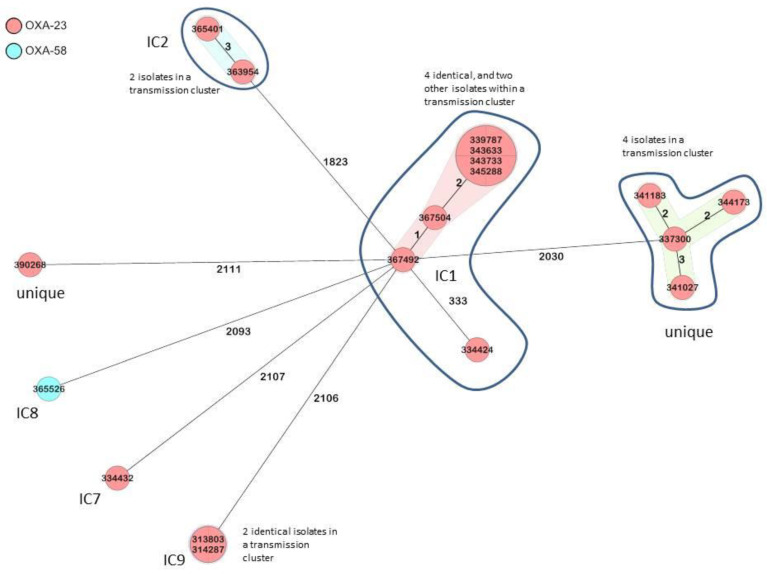
A minimum spanning tree of the *A. baumannii* strains based on 2390 target alleles (core genome) is presented. The study I.D.s of the isolates are shown within the nodes, and the numbers between the nodes indicate the number of alleles that were different. Isolates, which are elements of transmission clusters, are either shown within the same node or connected with transparent bands. Isolates are colored based on the detected *bla*_OXA_ gene.

**Table 1 microorganisms-09-02229-t001:** Summary of the isolates from Afghan patients based on their clonal lineages, multi-locus sequence types, intrinsic and acquired beta-lactamases, and history of acquisition.

Numeric I.D.	Clonal Lineage	Pasteur Sequence Type	Oxford Sequence Type	Acquired Beta-Lactamase	Intrinsic Beta-Lactamase	Year of Isolation	Anatomic Isolation Site
339787	IC1	94	1812/1813	*bla* _OXA-23_	*bla* _OXA-69_	2013	Lung
343633	IC1	94	1812/1813	*bla* _OXA-23_	*bla* _OXA-69_	2013	Bullet wound
343733	IC1	94	1812/1813	*bla* _OXA-23_	*bla* _OXA-69_	2013	Broncho-alveolar lavage
345288	IC1	94	1812/1813	*bla* _OXA-23_	*bla* _OXA-69_	2013	Unknown *
367504	IC1	94	1812/1813	*bla* _OXA-23_	*bla* _OXA-69_	2014	Rectum
367492	IC1	94	1812/1814	*bla* _OXA-23_	*bla* _OXA-69_	2014	Nose
334424	IC1	325	387	*bla* _OXA-23_	*bla* _OXA-69_	2013	Groin
363954	IC2	2	1114/1841	*bla* _OXA-23_	*bla* _OXA-66_	2014	Tracheal secretion
365401	IC2	2	1114/1841	*bla* _OXA-23_	*bla* _OXA-66_	2014	Nose
334432	IC7	25	2280	*bla* _OXA-23_	*bla* _OXA-64_	2013	Leg wound
365526	IC8	10	447	*bla* _OXA-58_	*bla* _OXA-68_	2018	Blood culture
313803	IC9	85	1752	*bla* _OXA-23_	*bla* _OXA-94_	2012	Unknown *
314287	IC9	85	1752	*bla* _OXA-23_	*bla* _OXA-94_	2012	Unknown *
337300	no	136	new	*bla* _OXA-23_	*bla* _OXA-317_	2013	Bullet wound (knee)
341183	no	136	new	*bla* _OXA-23_	*bla* _OXA-317_	2013	Thoracic wound
341027	no	136	new	*bla* _OXA-23_	*bla* _OXA-317_	2013	Blood culture
344173	no	136	new	*bla* _OXA-23_	*bla* _OXA-317_	2013	Unknown *
390268	no	52	931	*bla* _OXA-23_	*bla* _OXA-51_	2015	Unknown *

* Unknown = insufficient data available. I.D. = identity.

**Table 2 microorganisms-09-02229-t002:** Antimicrobial resistance determinants as identified within the *A. baumannii* strains.

Numeric I.D.	Beta-Lactamase Genes	Aminoglycoside Resistance Genes	Sulfonamide Resistance Genes	Macrolide Resistance Genes	Tetracycline Resistance Genes	Other Rarely Observed Resistance Genes
339787	*bla*_OXA-23_, *bla*_OXA-69__‡_	*aadB*-like, *aph*(*3*′)-*Ic*-like	*sul2*	*mph*(*E*), *msr*(*E*)		
343633	*bla*_OXA-23_, *bla*_OXA-69__‡_	*aadB*-like	*sul2*	*mph*(*E*), *msr*(*E*)		
343733	*bla*_OXA-23_, *bla*_OXA-69__‡_	*aadB*-like, *aph*(*3*′)-*Ic*	*sul2*	*mph*(*E*), *msr*(*E*)		
345288	*bla*_OXA-23_, *bla*_OXA-69__‡_	*aadB*-like, *aph*(*3*′)-*Ic*-like	*sul2*	*mph*(*E*), *msr*(*E*)		
367504	*bla*_OXA-23_, *bla*_OXA-69__‡_	*aadB*-*like, aph*(*3*′)-*Ic*	*sul2*	*mph*(*E*), *msr*(*E*)		
367492	*bla*_OXA-23_, *bla*_OXA-69__‡_	*aadB*-like, *aph*(*3*′)-*Ic*, *aph*(*3*′)-*VIa*-like	*sul2*	*mph*(*E*), *msr*(*E*)		
334424	*bla*_OXA-23_, *bla*_OXA-69__‡_	*aadB*-like, *aph*(*3*′)-*VIa*-like	*sul2*	*mph*(*E*), *msr*(*E*)		
363954	*bla*_OXA-23_, *bla*_OXA-66__‡_	*aac*(*3*)-*Ia*-like, *aadA1*, *strA*, *strB*	*sul1*, *sul2*		*tet*(*B*)-like	
365401	*bla*_OXA-23_, *bla*_OXA-66__‡_	*aac*(*3*)-*Ia*-like, *aadA1*, *aph*(*3*′)-*VIa*-like, *strA*, *strB*	*sul1*, *sul2*		*tet*(*B*)-like	
334432	*bla*_OXA-23_, *bla*_OXA-64__‡_, *bla*_PER-7_	*armA*, *strA*, *strB*	*sul1*, *sul2*	*mph*(*E*), *msr*(*E*)	*tet*(*B*)	*ARR*-*2* *, *cmlA1*-like **
365526	*bla*_CMY-4_, *bla*_OXA-58__‡_, *bla*_OXA-68_	*aac*(*3*)-*IId*-like, *aph*(*3*′)-*VIa*-like, *strA*, *strB*				
313803	*bla*_OXA-23_, *bla*_OXA-94__‡_	*aadB*-like, *aph*(*3*′)-*VIa*-like	*sul2*	*mph*(*E*), *msr*(*E*)		
314287	*bla*_OXA-23_, *bla*_OXA-94__‡_	*aadB*-like, *aph*(*3*′)-*VIa*-like	*sul2*	*mph*(*E*), *msr*(*E*)		
337300	*bla*_OXA-23_, *bla*_OXA-317__‡_	*aadB*-like, *aph*(*3*′)-*VIa*-like, *strA*, *strB*	*sul2*		*tet*(*39*)	
341183	*bla*_OXA-23_, *bla*_OXA-317__‡_	*aadB*-like, *aph*(*3*′)-*VIa*-like, *strA*, *strB*	*sul2*		*tet*(*39*)	
341027	*bla*_OXA-23_, *bla*_OXA-317__‡_	*aadB*-like, *aph*(*3*′)-*VIa*-like, *strA*, *strB*	*sul2*		*tet*(*39*)	
344173	*bla*_OXA-23_, *bla*_OXA-317__‡_	*aadB*-like, *aph*(*3*′)-*VIa*-like, *strA*, *strB*	*sul2*		*tet*(*39*)	
390268	*bla*_GES-11_, *bla*_OXA-23_, *bla*_OXA-51__‡_	*aph*(*3*′)-*VIa*-like, *aacA4*-like	*sul1*, *sul2*		*tet*(*B*)	*dfrA7* ***, *aac*(*6*′)*Ib*-*cr*-like ****

* mediating rifampicin resistance. ** mediating phenicol resistance. *** mediating trimethoprim resistance. **** mediating combined fluoroquinolone and aminoglycoside resistance. _‡_ intrinsic *bla*_OXA-51_ variant.

**Table 3 microorganisms-09-02229-t003:** Phenotypic resistance testing results based on microbroth dilution (applied for colistin) and automated resistance testing using a VITEK-II device (applied for all other assessed antimicrobial substances). MICs (minimum inhibitory concentrations) in µg/mL are shown in brackets. Interpretation was performed according to the EUCAST NAK (European Committee on Antimicrobial Susceptibility Testing, “Nationales Antibiotia-Sensitivitätstest-Kommittee”). S = susceptible. I = Susceptible at increased dosage. R = Resistant.

Numeric I.D.	Imipenem	Meropenem	Amikacin	Gentamicin	Tobramycin	Ciprofloxacin	Cotrimoxazole	Colistin
339787	R (8)	I (8)	S (4)	R (≥16)	R (≥16)	R (≥4)	R (≥320)	S (0.25)
343633	R (8)	R (≥16)	S (≤2)	R (≥16)	R (8)	R (≥4)	R (≥320)	S (0.5)
343733	R (8)	R (≥16)	S (4)	R (≥16)	R (8)	R (≥4)	R (≥320)	S (0.5)
345288	R (8)	R (≥ 6)	S (4)	R (≥16)	R (16)	R (≥4)	R (≥320)	S (0.25)
367504	R (≥16)	R (≥16)	S (4)	R (≥16)	R (≥16)	R (≥4)	R (≥320]	S (0.25)
367492	R (8)	R (≥16)	S (≤2)	R (≥16)	R (8)	R (≥4)	R (≥320)	S (0.5)
334424	R (8)	I (8)	R (16)	R (≥16)	R (8)	R (≥4)	R (≥320)	S (0.25)
363954	R (≥16)	R (≥16)	S (≤2)	R (8)	S (≤1)	R (≥4)	R (≥320)	S (0.5)
365401	R (≥16)	R (≥16)	R (16)	R (8)	S (≤1)	R (≥4)	R (≥320)	S (0.5)
334432	R (≥16)	R (≥16)	R (≥64)	R (8)	R (≥16)	R (≥4)	R (>320)	S (0.5)
365526	R (≥16)	R (≥16)	R (≥64)	R (8)	R (8)	R (≥4)	R (≥320)	S (0.5)
313803	R (≥16)	R (≥16)	R (16)	R (≥16)	R (8)	R (≥4)	R (≥320)	S (0.25)
314287	R (≥16)	R (≥16)	R (16)	R (≥16)	R (8)	R (≥4)	R (160)	S (0.5)
337300	R (≥16)	R (≥16)	R (≥64)	R (≥16)	R (≥16)	R (≥4)	R (≥320)	S (0.5)
341183	R (≥16)	I (8)	R (≥64)	R (≥16)	R (≥16)	R (≥4)	R (≥320)	S (0.5)
341027	R (≥16)	R (≥16)	R (≥64)	R (≥16)	R (≥16)	R (≥4)	R (160)	S (0.5)
344173	R (≥16)	R (≥16)	R (≥64)	R (≥16)	R (≥16)	R (≥4)	R (160)	S (0.5)
390268	R (≥16)	R (≥16)	R (≥64)	R (8)	R (8)	R (≥4)	R (≥320)	S (1.0)

## Data Availability

All relevant data are provided in the manuscript or via the links provided in the manuscript. Raw data may be made accessible upon reasonable request.

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
