# Peer review of "Molecular Epidemiology of Carbapenem-Resistant Acinetobacter baumannii Strains Isolated at the German Military Field Laboratory in Mazar-e Sharif, Afghanistan"

_microorganisms, 2021, doi:10.3390/microorganisms9112229_

Round 1

Reviewer 1 Report

In this manuscript, Higgins et al., elucidates the phenotypic and genetypic characterestics of multidrug resistant A. baumannii isolates from a military medical service in Afghanistan. They denote the predominance of blaOXA-23  in carbapenem resistant isolates. Also, the authors nicely review literature on A. baumannii's implication in skin and soft tissue infections. 

Ubiquitous susceptiblity to colistin is quite interesting, whereas, data on tigecycline efficacy could be added, if they are available.

The manuscript is well written and presented data are quite interesting and well documented

Author Response

Reviewer 1, point 1:

Ubiquitous susceptiblity to colistin is quite interesting, whereas, data on tigecycline efficacy could be added, if they are available.

Authors:

In the last sentence of the new fourth paragraph of the discussion, we have stated that tigecycline was not assessed, as susceptibility testing results were interpreted in line with EUCAST (European Committee on Antimicrobial Susceptibility Testing) standards which do not describe breakpoints for tigecycline.

English language and style were checked again as suggested by both reviewers.

Reviewer 2 Report

The authors took a picture on A. baumannii strains circulating in Afghanistan from 2013 to 2015, with one isolate in 2018 and their antibiotic-resistance genes.

The paper is well written and presented. Due to the limitation, no speculations can be made about the evolution of these strains.

A big concern, with no explanation in the text, is why so many strains were susceptible to amikacin even possessing a number of Aminoglycoside resistance genes?

What about insertion sequence, transposons, and phage sequences?

Minor revision: in the first mention of Acinetobacter baumannii, there is no need do explain Acinetobacter (A.). You can use A. baumannii directly throughtout the text.

Author Response

Reviewer 2, point 1:

A big concern, with no explanation in the text, is why so many strains were susceptible to amikacin even possessing a number of Aminoglycoside resistance genes?

Authors:

Not all the aminoglycoside modifying enzymes we detected confer amikacin resistance. From the amikacin susceptible isolates, two have aph(3’)-VIa but are not amikacin resistant, while all the amikacin resistant isolates have aph(3’)-Via except one which has another amikacin resistance gene, armA. We have addressed the issue of imperfect matching of the detection of aminoglycoside resistance genes and phenotypically measured minimum inhibitory concentrations in the new fourth paragraph of the discussion. Further details are provided in the results section, subheading “3.2. Genotypic and phenotypic resistance” by the newly added fourth and fifth sentences.

Reviewer 2, point 2:

What about insertion sequence, transposons, and phage sequences?

Authors:

This is a good point from the reviewer. Given that all but one isolate had blaOXA-23, they will all have copies of ISAba1 as this is part of the OXA-23 mobilization and expression. The OXA-58-expressing isolate will have ISAba2 and/or ISAba3 as this is associated with blaOXA-58. These IS elements are present in multiple copies and the reads pile up into single contigs. The only way to resolve this is to use long read sequencing which we did not perform for reasons of funding restriction. Thus, we know that they are there, but we cannot determine easily where they are in the genome. We have addressed this problem as a new fourth limitation at the end of the discussion paragraph.

Reviewer 2, point 3:

Minor revision: in the first mention of Acinetobacter baumannii, there is no need do explain Acinetobacter (A.). You can use A. baumannii directly throughtout the text.

Authors:

The “A” in brackets has been removed as suggested.

English language and style were checked again as suggested by both reviewers.